# Methodology of Drought Stress Research: Experimental Setup and Physiological Characterization

**DOI:** 10.3390/ijms19124089

**Published:** 2018-12-17

**Authors:** Natalia Osmolovskaya, Julia Shumilina, Ahyoung Kim, Anna Didio, Tatiana Grishina, Tatiana Bilova, Olga A. Keltsieva, Vladimir Zhukov, Igor Tikhonovich, Elena Tarakhovskaya, Andrej Frolov, Ludger A. Wessjohann

**Affiliations:** 1Department of Plant Physiology and Biochemistry, St. Petersburg State University, St. Petersburg 199034, Russia; natalia_osm@mail.ru (N.O.); bilova.tatiana@gmail.com (T.B.); elena.tarakhovskaya@gmail.com (E.T.); 2Department of Biochemistry, St. Petersburg State University, St. Petersburg 199904, Russia; schumilina.u@yandex.ru (J.S.); didio1992@yandex.ru (A.D.); tgrishina@mail.ru (T.G.); 3Department of Bioorganic Chemistry, Leibniz Institute of Plant Biochemistry, Halle (Saale) 06120, Germany; ariyong1002@gmail.com; 4Institute of Analytical Instrumentation, Russian Academy of Science, St. Petersburg 190103, Russia; keltcieva@gmail.com; 5All-Russia Research Institute for Agricultural Microbiology, St. Petersburg 196608, Russia; vladimir.zhukoff@gmail.com (V.Z.); arriam2008@yandex.ru (I.T.); 6Department of Genetics and Biotechnology, St. Petersburg State University, St. Petersburg 199034, Russia; 7Department of Scientific Information, Russian Academy of Sciences Library, St. Petersburg 199034, Russia

**Keywords:** drought stress, drought models, drought tolerance, oxidative stress, phytohormones, polyethylene glycol (PEG), stress markers

## Abstract

Drought is one of the major stress factors affecting the growth and development of plants. In this context, drought-related losses of crop plant productivity impede sustainable agriculture all over the world. In general, plants respond to water deficits by multiple physiological and metabolic adaptations at the molecular, cellular, and organism levels. To understand the underlying mechanisms of drought tolerance, adequate stress models and arrays of reliable stress markers are required. Therefore, in this review we comprehensively address currently available models of drought stress, based on culturing plants in soil, hydroponically, or in agar culture, and critically discuss advantages and limitations of each design. We also address the methodology of drought stress characterization and discuss it in the context of real experimental approaches. Further, we highlight the trends of methodological developments in drought stress research, i.e., complementing conventional tests with quantification of phytohormones and reactive oxygen species (ROS), measuring antioxidant enzyme activities, and comprehensively profiling transcriptome, proteome, and metabolome.

## 1. Introduction

Being a natural climatic feature, drought occurs in almost all climate zones with varying frequency, severity, and duration, and is one of the most deleterious factors of environmental stress [1,2]. Indeed, even a short-term water deficit results in essential annual losses of crop yields [3,4], impeding sustainable agriculture all over the world [5,6,7]. Due to oncoming climate changes, the frequency and duration of drought periods will increase, making this factor one of the most important threats of the current century [8,9].

In the context of agriculture, drought is defined as a period of below-average precipitation [10], when the amounts of available water in the plant rhizosphere drop below the limits required for efficient growth and biomass production [11]. Such a soil water deficit can be persistent in climate zones characterized by low water availability, or by intermittent and unpredictable water supply during the vegetative period [12]. Because of this, drought is the major environmental stressor, affecting the plant’s growth and development by disrupting its water status [13]. This dramatically affects all key physiological processes, such as photosynthesis, respiration, and uptake of mineral nutrients [14,15]. First, drought compromises stomata function, impairs gas exchange, and leads to overproduction of reactive oxygen species (ROS) and development of oxidative stress [16]. Second, water deficit inhibits cell division, expansion of leaf surface, growth of stem, and proliferation of root cells [7]. In concert, all these factors dramatically reduce plant productivity and might lead to the death of drought-sensitive plants upon prolonged exposure to drought [17].

At the quantitative level, water deficit in the environment can be characterized by a decrease of soil water potential (Ψ_w_) [18]. According to the van’t Hoff equation, it indicates a decrease in free energy of substrate water that makes water uptake from the medium under these conditions thermodynamically unfavorable and loss of water by the plant more probable. Values of Ψ_w_ from 0 to −0.3 MPa are characteristic for well-watered plants, whereas values below −0.4 MPa correspond to moderate water stress, and potentials of −1.5 to −2.0 MPa represent severe stress and permanent loss of turgor [19]. However, these values vary among species and drought models. They are based on experience with seeds and seedlings, which are commonly more drought tolerant. Thus, in our experience, Ψ_w_ values of −0.3 to −0.8 MPaare more typical for experimentally useful, i.e., recoverable, moderate drought stress in plants beyond the seedling stage (v.i.). In general, leaf Ψ_w_ can be determined by several approaches. In the easiest but most reliable way, Ψ_w_ can be addressed by the gravimetric method [20]. It can also be accomplished with a Scholander pressure chamber and thermocouple psychrometer [21] or tensiometer [22]. Thermocouple psychrometry is one of the most popular methods, and is usually accomplished with press saps or freeze-thawed leaf disks [23]. Recently, a new method was proposed for determination of Ψ_w_ in leaf cell apoplast, relying on the measurement of photosynthetic CO_2_/H_2_O gas exchange [24].

It is important to mention that not only the degree of Ψ_w_ decrease, but also its duration, can affect the plant organism [25]. Therefore, water stress often develops upon minimal reduction of soil Ψ_w_. To avoid this scenario, plants adopt various strategies to prevent water loss, to preserve water supply even under reduced Ψ_w_, and to sustain periods of unfavorable water regimen accompanied by low water content in tissues [10]. These drought-induced alterations can affect plant morphology, physiology, and biochemistry in degree -depending on plant species, developmental stage, and duration and severity of drought [4,6,7,26,27].

The main strategies employed by plants to sustain water deficit are (i) drought escape, (ii) drought avoidance, and (iii) drought tolerance [28]. Generally, all three strategies impact the development of the state known as drought resistance, which can be defined as the ability to maintain favorable water balance and turgidity under drought conditions. In the escape strategy, plants complete their life or growth cycle before the impact of drought causes harm, i.e., they use a seasonal response [4]. The strategy of drought avoidance relies on enhanced water uptake and reduced water loss, whereas drought tolerance is mediated by osmotic adjustment, extension of antioxidant capacity, and development of desiccation tolerance [28]. On one hand, these strategies represent different steps of drought response (Figure 1). On the other hand, they might indicate different climatic and ecological specializations of plant species [29]. This concept of stress avoidance and stress tolerance proposed by Levitt [30] provides insight into plant responses to a relevant decrease of Ψ_w_ at the cell and organism levels [10].

As can be seen from Figure 1, the first response of the plant organism to drought as a drought-resistance strategy relies on avoiding water deficit [31] by maintaining tissue Ψ_w_ by increasing water uptake or restricting water loss [32]. At the early steps of drought response, it is mainly achieved by stomata closure, triggered by abscisic acid (ABA). However, according to Muller et al. [33], the rapid expansion of roots and young leaves (as a major C sink) is affected earlier and more intensely than photosynthesis (C source); accordingly, root growth is enhanced to provide sufficient water uptake under drought conditions. These avoidance mechanisms can secure the maintenance of crop plant productivity during short-term periods of water deficiency [18]. However, this is achieved at the price of reduced CO_2_ uptake, a dramatic drop in photosynthesis rate, and redirection of assimilate transport for enhancement of root growth [33,34]. When drought persists for a long time and adaptive capacities of the avoidance strategy are not sufficient to sustain plant growth and productivity, other mechanisms might be involved. At this step, mechanisms such as accumulation of compatible solutes and protective proteins (so-called metabolic adjustment), cell wall hardening, ROS detoxification, and metabolic changes are involved in establishing drought tolerance [10].

Thus, plant drought resistance is a complex process that requires a global view to understand its underlying mechanisms. Obviously, the majority of molecular events triggered by a decrease of tissue Ψ_w_ cannot be unambiguously attributed solely to avoidance or tolerance strategy. Therefore, a complex multilevel regulatory network controlling plant adaptive responses to drought stress is required. Studies of responses to water deficit such as stomata closure, expression of stress-specific genes, accumulation of osmolytes, and up regulation of antioxidant systems recently made considerable progress [17,35,36,37,38]. It was shown that the mechanisms underlying stress resistance are crucial for plant survival and are associated with significant changes in the patterns of metabolites and proteins [10,15,35]. Hence, analyzing the changes in plant metabolome and proteome associated with the onset of drought might be an important step in breeding and engineering plants with increased drought resistance [15,35] or developing plant protectants against drought stress [39].

Recently, Wang et al. [15] comprehensively reviewed drought-related effects on the plant proteome, including changes in signal reception and transduction, ROS scavenging, osmotic regulation, protein synthesis/turnover, modulation of cell structure, and carbohydrate and energy metabolism. These functional patterns of plant response to drought gave access to understanding of fine mechanisms underlying the process of stress tolerance. Apparently, for successful study of plant responses to drought stress under experimental conditions, reliable and adequate stress models are required. Accordingly, various drought models have been established (Table 1). However, the available information is often complex, incomplete, and inconsistent. A comprehensive literature search for drought tolerance research shows great variability and inconsistency in the experimental designs and methods for stress characterization [15]. Therefore, here we systematically address different experimental setups for establishing drought stress models and consider physiological and biochemical methods for their characterization.

## 2. Experimental Models of Drought Stress

Despite a large variety of available drought models, according to their basic setup, all of these techniques can be classified as soil-based, aqueous culture–based, or agar-based. The common feature of all drought stress models is reduction of the water potential in the substrate or medium surrounding plant roots. However, individual methods have different applicability limitations and vary in terms of the scientific questions they can address. Therefore, the advantages and disadvantages of each model need to be carefully considered prior to experiment planning.

### 2.1. Soil-Based Drought Models

The obvious advantage of this model strategy is the close similarity of experimental conditions to actual drought in nature and agriculture. In this case, the decrease of soil Ψ_w_ is established by gradual decline or immediate interruption of plant watering [40]. Such models adequately simulate short-term drought, which represents the most frequent case in the European agricultural practice due to varying weather conditions [41,42]. However, difficulty controlling the substrate Ψ_w_ represents an essential limitation of this approach. Indeed, in this experimental setup, the severity of drought stress is determined by the rates of water evaporation from the soil surface and consumption by the plant [43]. As these cannot be defined by the researcher and depend on multiple factors, reproducibility and predictability of such experiments are always questionable. Moreover, as the rates of water consumption and evaporation are relatively high, this model does not allow probing long-term drought responses, such as accumulation of osmoprotective metabolites or proteins and cell wall modifications [10]. Therefore, many important aspects of plant drought tolerance and adaptation to low Ψ_w_, such as, for example, accumulation of osmoprotective proteins and hardening of cell walls, can be overlooked in this experimental setup, although using large and deep pots might improve this situation [10].

Despite the above-mentioned problems, several improvements can be made to increase the reproducibility and reliability of soil models. First of all, in this type of experiment, the size and structure of soil particles, as well as their water capacity, should be taken into account. Thus, to achieve moderate (i.e., less severe) drought conditions, in an optimized variant of this model, plants are grown in foil-sealed vessels to prevent water evaporation from the soil surface [5]. Thereby, each pot can be equipped with a piece of tubing inserted into the soil to facilitate rewatering of plants. Due to the water supply, in this model, water deficit can be increased gradually, making it possible to address long-term plant responses to drought [44]. Moreover, stability of the water regimen can be improved by an automated irrigation system.

Recently, Todaka et al. [40] introduced an automatic irrigation system to relay monitoring of actual water content in soil. Using this approach, the authors proposed a drought model able to ensure the desired values of Ψ_w_ (−9.8, −31.0, and −309.9 KPa). However, this system failed to reproduce the conditions of severe dehydration. Although the optimized method described above is reliable and reproducible enough, repeated measurements of leaf and soil Ψ_w_ are laborious and require large amounts of plant material, which are hardly available in long-term experiments under reproducible laboratory conditions. For example, such a restriction can be critical when mutants or transgenic plants are dealt with, in particular those with reduced stomata density or small leaf area [45].

An elegant way to avoid this complication is to culture mutant or transgenic plants in the same pot with the reference plants, e.g., the wild-type (wt) counterparts [10]. In this case, leaf Ψ_w_ determination can be limited to the reference (or wt) plants, which are commonly more suitable for assessing stress markers. The obtained results can be extrapolated to the mutants. In this case, both reference or wt and experimental or mutant plants would grow in the same medium and therefore be exposed to the same soil Ψ_w_ if they are planted in a suitable scheme and position. The best way to provide a quantitative characteristic of drought stress by this approach is to complement it with a measurement of soil Ψ_w_ at the end of the dehydration period. Analogously, this method can be applied to untreated and treated plants in assays for chemical drought tolerance enhancers or other phytoeffectors (v.i.) to be tested.

It is important to mention the setups that rely on inert substrate, such as vermiculite or perlite, as soil substitutes. The advantage of this approach is that the roots of experimental plants can be pulled out easily and without damage to investigate drought-related changes in water potential [46] or oxidative and metabolic responses [47] at the root level. Inert substrates are suitable for studying the effects of drought in legume–rhizobial nodule symbiosis [48]. On the other hand, the certain disadvantage is that watering, unlike soil culture, is carried out not with water, but with a nutrient solution, so the impact of drought by cessation of watering is accompanied by the appearance of another stress factor, i.e., a deficiency of mineral elements.

### 2.2. Drought Models Based on Hydroponic Aqueous Culture

Despite the high relevance of soil-based drought models because of their similarity to natural conditions, they all have a common intrinsic limitation: the difficulty of adequately controlling Ψ_w_ in the root microenvironment. However, this is critical when a precise definition of substrate Ψ_w_ is required, as in multiple or long-term experiments comparable over months (and seasons). Therefore, the models, based on aqueous hydroponic culture with predictably decreased Ψ_w_ of nutritional solution, might be advantageous for such applications. The easiest way to reduce the Ψ_w_ of growth medium is to decrease its level in pots and partially exposure roots to air, as was shown for lettuce by Koyama et al. [49]. To simulate severe dehydration, plant roots can be left under air for up to eight hours [50], allowing the severity of simulated drought to be defined by the duration and repetitions of the dehydration process. This approach is based on the fact that Ψ_w_ of leaves, at least to some extent, corresponds to the index of water availability for plants, which in turn depends on water potentials of soil and plant roots [51]. Thus, experimentally affecting the Ψ_w_ of roots influences the Ψ_w_ of leaves as well. When using this approach, however, one needs to keep in mind that the degree of dehydration and kinetics would strongly depend on air humidity. Further, it is important to remember that in this case the plant response is dependent on root distribution (e.g., long vs. short roots).

Despite the ease of the above approach, most often desired Ψ_w_ values of plant rhizosphere are obtained by supplementing nutrient solutions with osmotically active substances (osmolytes, which reduce available water), taken in calculated concentrations. This approach is based on the simulation of drought by application of osmotic stress, i.e., increasing the medium osmotic pressure compared to that of plant tissues [52]. Similar events occur in soil when the water content decreases (due to evaporation and absorption by the plant) and the concentrations of solutes grow, resulting in an increased osmotic component of the water potential [53]. Thus, the described setup corresponds well with natural drought. This strategy allows precise adjustment of Ψ_w_ and efficient monitoring of its magnitude, resulting in high accuracy, reproducibility, and interexperimental comparability of acquired data [54]. However, when working with this kind of drought model, selecting an appropriate osmolyte requires special attention. Thus, low-molecular-weight osmolytes (e.g., sugar alcohols and sodium chloride) routinely used in early studies [55] demonstrate strong negative side effects when applied in experimental drought. Indeed, these compounds easily penetrate cell walls and plasma membranes, increasing intracellular osmotic pressure and leading to plasmolysis [56]. Any salts also change ion titers and distribution in plants, affect ionic strength, and trigger the process of ion transport. On the other hand, nonionic carbohydrate-related osmolytes (e.g., sorbitol and mannitol) are readily involved in cellular metabolism themselves, and thus might directly affect the results of the experiment [56], as they are often toxic to plants [57]. They can also increase mold growth under commonly nonsterile conditions. Because of this, the use of biologically inert polymeric osmolytes is preferable and advantageous [58]. Therefore, currently, drought stress models rely on presumably nonpermeable high-molecular-weight osmolyte polyethylene glycol (PEG) with an average molecular weight of 6000 Da or more [55,59].

It is well documented that PEG effectively decreases medium Ψ_w_, thereby disrupting absorption of water by plant roots [60]. In terms of this approach, 5–20% (*w*/*v*) [61] or even 40% (*w*/*v*) [62] PEG in growth medium enables a stabile decrease of Ψ_w_ during any desired period of time [63]. Importantly, PEG-based aqueous models allow the setup of recovery experiments by transfer of stressed plants to PEG-free nutrient solution or exchange of the PEG solution [10]. Therefore, PEG-based models of drought stress represent the method of choice in molecular biology and plant protectant studies and screening experiments [64]. One issue yet underexplored in the PEG model is the complexing ability of PEGs on metal ion species and thus the altered availability of the various ions for the plant. However, also under drought conditions, ion availability eventually changes and decreases.

One of the most promising applications of aqueous PEG-based models of osmotic stress is screening for potential drought-protective compounds. Substances that influence plant performance (without being plant protectants against biotic stress, e.g., from pathogens) in agrochemistry are defined as phytoeffectors and include drought stress tolerance enhancers. Phytoeffectors are able to prime crop plants against short-term drought and ensure that their productivity is sustained under drought conditions with spatiotemporal control, largely independent of the crop species or variety used. Such effects were described for salicylic acid and its derivatives [65], as well as for various fungicides of the triazole [66] and imidacloprid [67,68] families. The drought-protective effects of small molecules on a plant organism are usually mediated by inhibition of enzymes, involved in plant response to stress, as was described for poly(ADP-ribose) polymerase (PARP)in the beginning of this decade [68], although later at least direct involvement of PARP appeared doubtful [69]. If a molecular target for drought stress effects is known, and ideally the active site too, methods of computational chemistry like virtual screening and molecular docking approaches [70] allow virtual screening of thousands of structures with millions of conformers. The most promising candidates for wet lab testing can thus be identified.

For rapid screening of such compounds, a reliable model based on a *Lemna minor* culture was recently developed in our group [68]. This technique (Figure 2A) relies on a microtiter plate format and assumes treatment of plants with PEG6000 or PEG8000 supplementing the growth medium in the presence and absence of potential phytoeffectors. After a 24 h stress period, plants are transferred to a PEG-free medium, and stress recovery is monitored for further 48 h, before the protective effect is assessed by attenuation of growth inhibition via measurement of leaf peak area increase by means of a 2D-photodocumentation visualization system.

The *Lemna* system has several advantages over classical spraying systems: Plants are all clones, reproducing by budding, and they are small and can be grown in microtiter plates (6-, 12-, or 24-well format) under sterile conditions. The small scale allows medium-throughput screening with small amounts of compounds. Most importantly, these can be applied in a concentration-dependent manner to the multiwell plate well (while spraying or dumping delivers only uncertain amounts to plants), and both root and leaf uptake is ensured. The leaves are flat and 2D phenotyping is easily done with the respective software [68]. For better reproducibility, initial root length should be unified, and until termination of the experiments, plant growth should not be limited by well size.

Despite their wide use, PEG-based models have some intrinsic limitations that need to be taken into account when planning experiments [63]. First, PEG-containing nutrient solutions are characterized by high viscosity, which compromises diffusion of oxygen to the roots, especially in deeper vessels, and can cause hypoxia [10]. To prevent the development of hypoxia, additional aeration needs to be provided for plants grown in PEG-containing medium. For this, air is continuously supplied by pumps through silicone tubes connected to the culture vessels [71]. Although this approach can be easily established for larger plants (as it was done in our lab with *B. napus*; Figure 2B [72]), small model plants like Arabidopsis, typically grown in small vessels on large scale for highly replicated biological experiments, cannot be supplied with air individually and are typically grown under hypoxic conditions [73]. Small and flat vessels like the wells used in the *Lemna* system [68] are usually not prone to such problems.

Another possible issue is absorption and accumulation of PEG with molecular weight 4000–8000 Da in plant roots, which might result in damage [74]. The accompanying partial root dysfunction might impact leaf dehydration in an unpredictable way. Thus, stress responses observed in plant shoots are only partly related to osmotic stress applied by PEG solution. The impact of PEG-related root damage on these responses is difficult to estimate, but is obviously increased when plant transfer on PEG-containing medium is accompanied with wounding of roots, which should be avoided [75].

### 2.3. Agar-Based Drought Models

In general, growing plants in agar allows avoiding or reducing development of the hypoxic state. As this is especially relevant for Arabidopsis, agar-based models are widely used in plant biology, and specifically in drought stress experiments with *Arabidopsis thaliana* seedlings [76]. Thus, van der Weele et al. proposed an agar-based PEG infusion model relying on saturation of solidified agar (filled in Petri plates) with Murashige and Skoog medium supplemented with PEG8000 during two days [77]. Unfortunately, PEG affects the solidification of agar, therefore adding it directly to the agar medium under preparation is not advisable [73]. Because of this, generating a desired Ψ_w_ of agar medium is achieved by diffusing PEG from a concentrated overlay solution into preformed, solidified agar. Adjusting the concentration of the overlay solution, the equilibrium in Ψ_w_ between aqueous overlay medium and agar is achieved after 24 h of diffusion [76]. After decantating the PEG solution, seedlings can be transferred to the now PEG-containing agar medium (stress application), and eventually plants can be replanted to a PEG-free one after a defined treatment period (recovery).

Due to a constant character of Ψ_w_, the agar-based PEG infusion model is advantageous compared to those based on soil or (nonaerated) aqueous culture. Thus, the Ψ_w_ of seedlings can achieve equilibrium with the agar medium during treatment time. Under soil drying conditions, this is impossible, as soil Ψ_w_ changes continuously along with water evaporation and consumption by the plant. On the other hand, due to PEG interfering with root integrity [78], this equilibrium is also hardly achievable in aqueous PEG solutions (especially when PEG concentration is high). Thus, the agar-based model system currently is an ideal choice to address dehydration avoidance and mechanisms of dehydration tolerance [10]. Most commonly, PEG concentrations in the agar medium do not exceed the values needed for medium to medium-high drought stress, i.e., Ψ_w_ < −1.2 MPa [57]. However, based on the solubility of PEG8000 in water, the agar-based infusion model can be established in a broad range of overlay medium Ψ_w_ values from −0.47 MPa to −3.02 MPa [79].

The agar-based PEG infusion model was successfully applied to different plants and fungi [75]. Further, a similar setup (10% *w*/*v* PEG6000 in the overlay medium) was used to probe the effect of water stress on rape oilseed (*Brassica napus*) germination and seedling development [80,81]. An essential limitation of the setup, originally proposed by Verslues and co-workers [10], was its applicability to only the early steps of plant ontogenesis, seed germination and seedling development. Thus, this method was inapplicable to mature plants, and corresponding stress responses characteristic of later stages of ontogenesis could not be addressed.

Therefore, to extend the agar-based approach to mature organisms, we modified the method of Verslues and co-workers to 5-to-7-week-old *A. thaliana* plants [35]. This setup combined germination on agar in truncated polypropylene tubes, growth for 4–5 weeks in aqueous culture, and transfer to agar medium preinfused with PEG8000 solution with a polymer concentration corresponding to the targeted substrate Ψ_w_ (Figure 2C).

In general, our observations confirmed earlier data indicating higher sensitivity of mature plants to drought compared to seedlings [10], although stress tolerance varies essentially between species. Thus, in contrast to seedlings, application of Ψ_w_ below −0.6 MPa led to reduced survival of plants over a period of 7 days, whereas a drop to −0.4 MPa was accompanied by significant alterations in plant metabolome and proteome, indicating metabolic adjustment and changes in redox metabolism [35]. Soybean turned to be more resistant to osmotic stress applied in an agar-based model and successfully survived osmotic stress applied by 8 and 16 (*w*/*v*) PEG for two weeks for pre- and post-flowering treatments [82].

To summarize, compared to other setups, the agar-based PEG infusion model has two fundamental advantages. First, it provides a stable and reproducible decrease of substrate Ψ_w_ that cannot be achieved with the soil-based model. On the other hand, compared to models based on aqueous culture, it has higher relevance to the conditions of a real field, as it relies on a solid substrate. Second, the agar-based model allows precise Ψ_w_ setting in plant rhizosphere without accompanying hypoxia and PEG-related root toxicity. It is important to keep in mind, however, that this setup does not allow a direct extrapolation of drought effects to the field or ecosystem due to the model’s simplicity, which doesnot consider water gradient in soil and heterogeneity in terms of water holding capacity. For fast (pre-)screening of phytoeffectors, especially if only small amounts of test compounds are available, the *Lemna minor* aqueous system has advantages [68] but must be complemented later by the solid medium method for validation [35].

## 3. Physiological and Biochemical Characterization of Drought Stress

Adequate and correct application of an experimental drought stress model requires comprehensive characterization at the levels of physiology, biochemistry, and molecular biology. These experiments deliver objective information on the actual functional state of the plant organism and its metabolic response to stress. This block of data is necessary to confirm the stressed state of experimental plants (i.e., development of stress response) and to estimate the severity of stress-related alterations. Accordingly, a panel of physiological and biochemical markers of drought stress ideally accompanies any study relying on modeling setups. Importantly, these markers can be used for dynamic characterization of plant adaptive responses throughout the whole experiment, i.e., acquisition of stress kinetics (Table 2). Thus, ideally, selection of the markers needs to consider all steps of drought response, starting from drought perception. It is assumed that drought is recognized by roots, which send a chemical message to the shoot [83]. Abscisic acid (ABA) plays a key role in this signaling [84]. This effector is synthesized in response to hydraulic signals in vascular tissues and further transported to leaf epidermis cells. Resulting stomata closure results in suppression of xylem transport, decreased turgor, and root growth arrest [37].

### 3.1. Water Status and Photosynthetic Parameters as Markers of Drought Stress

One of the first detectable symptoms of drought is dehydration of plant tissues, which is characterized by a decrease of Ψ_w_ and loss of leaf turgor [6]. Due to its simplicity, low time expense, and robustness, the measurement of leaf water potential prior to sunrise is one of the most commonly used tests for this marker [85]. Although critical values of tissue water potentials are species-specific, Ψ_w_ of less than −0.8 MPa is commonly recognized as a sign of drought stress [86]. On the other hand, the degree of water loss can be reliably assessed by a decrease of leaf relative water content (LRWC) [87]. In the easiest and most straightforward way, this parameter can be addressed by the gravimetric method and calculating dry weight/fresh weight ratios [88]. Despite its simplicity, this approach yields highly reproducible data. An obvious disadvantage of this method is its destructive character, i.e., consumption of plant material for each determination [85]. In this context, a nondestructive technology based on automatic assessment of short-wave infrared irradiation reflected from the leaf surface might be a good alternative [85]. Another nondestructive approach relies on long-term phytomonitoring, i.e., continuous measurement of leaf transpiration, turgor, and xylem flow by means of nondamaging sensors attached to the plant [89].

One of the primary plant responses to dehydration is stomata closure, which is intended toprevent transpiration-related water loss, and is essential for the success of the drought avoidance strategy [90]. Similarly to dehydration itself, this parameter can be quantitatively characterized [91]. Experimentally, it can be done by measuring the rate of gas flow through a leaf surface or the electrical conductivity of the water film (of constant ionic strength) on the leaf surface [92]. Therefore, stomata conductance is usually expressed in mmol/m^2^/s [92]. Technically, such experiments are based on porometric measurements, i.e., determining times required for increased air humidity in an isolated chamber with a leaf inside [93].

Since stomata closure disrupts the supply of parenchyma cells with carbon dioxide, drought ultimately negatively affects the efficiency of photosynthesis by inhibiting carbon assimilation and light reactions [5]. In the simplest way, photosynthetic activity can be addressed by quantitative determination of pigments: chlorophylls (at least chlorophyll a) and carotenoids [94]. Thereby, decreased chlorophyll level is considered a symptom of oxidative stress and may be the result of pigment photo-oxidation and chlorophyll degradation [95]. Accordingly, as was shown in a comparative screening of barley genotypes, higher chlorophyll content was generally associated with higher drought tolerance [94,96]. This allows us to consider this indicator as an important marker of plant functional state under drought conditions.

Besides degradation of photosynthetic pigments, dehydration negatively affects the whole photosynthetic apparatus [97]. One of the most reliable markers of this process is decreased photosystem II (PS II) activity [98]. Both relative chlorophyll content and PS II efficiency can be easily quantified with pulse amplitude modulation (PAM) fluorometry [99,100]. Thereby, the ratio of minimum (background) and potentially maximum chlorophyll fluorescence (Fv/Fm) is interpreted as the maximum of PS II photochemical activity and might be considered as a reliable marker of PS II photoinhibition and one of the most important indicators of drought stress [101]. Importantly, the chlorophyll fluorescence is registered in vivo, thus it does not require sampling of plant material [102]. Interestingly, in some cases, drought does not cause any alterations of PS II activity. This result, observed with potato leaves, can be explained by photochemical quenching of excess light energy by increased photorespiration [103]. It needs to be taken into account that besides drought stress, the onset of senescence can underlie a decrease in Fv/Fm ratio [104].

In agreement with the described mechanisms, the features protecting the chloroplast photosynthetic machinery from oxidative damage might increase stress tolerance. This was illustrated in a comparative study of two *B. napus* cultivars grown for 3 weeks in aerated aqueous nutrient medium with Ψ_w_ of −0.6 MPa (18% *w*/*v* PEG8000) [105]. The developing stress could be recognized in both cultivars by a pronounced decrease in growth and photosynthetic parameters, including PS II activity and chlorophyll a content. However, the cultivar with higher leaf contents of chlorophyll a and carotenoids, as well as higher Fv/Fm ratios, demonstrated a clearly higher drought tolerance. Thus, it could be concluded that the quantum yield of photosynthesis and the chlorophyll a content could be effective selection criteria in screening for cultivars of crop plants with drought tolerance [105].

### 3.2. Changes in Phytohormone Patterns as Markers of Drought Stress

Plant response to environmental stress is a complex process, precisely tuned by multiple regulatory systems [12,106]. In particular, dehydration triggers activation of signal transduction cascades, including long-distance transport steps mediated by phytohormones [107]. Specifically, drought-induced stomata closure is regulated by abscisic acid (ABA) and relies on ABA-dependent signaling pathways [108]. Upon dehydration, ABA tissue content in Arabidopsis leaves can be increased up to 30-fold [107]. In a time-course study of the drought-avoidance response performed with Arabidopsis, early accumulation of ABA and induction of associated signaling genes coincided with a decrease in stomata conductance, as revealed by a panel of physiological, biochemical, and molecular biology methods [12]. Therefore, increased ABA in leaf cells represents a reliable marker of drought stress in model experiments [35].

Besides ABA, several other hormones and their interaction networks have an impact on the control of stomata conductance during water deficit. Thus, auxins, cytokinins, and ethylene are prone to inhibit the ABA-mediated stomata closure mechanism, whereas brassinosteroids, isoleucinyl jasmonates, and jasmonic and salicylic acids support the effects of ABA [109]. Jasmonic acid and its derivatives play a significant role in plant response to drought in terms of opening and closing of stomata [110], acting in an interplay with ABA and starting ABA signaling transduction [111]. In contrast to jasmonates and ABA, ethylene is involved in the stimulation of stomata opening via inhibition of nicotinamide adenine dinucleotide phosphate, reduced form (NADPH) oxidase in the leaves of plants, responsible for the launch of ROS-dependent stomata closure pathways [112], but ethylene also induces senescence. Thus, despite their impact on drought response, the mentioned phytohormones have complex patterns of effects [107]. Therefore, their use as drought stress markers is hardly possible. Similarly, their potential to be applied as phytoeffectors in the field is limited. Apart from cost, bioavailability, and stability issues, it would require an extremely balanced mixture of suitable hormones, adapted in each case to the plant species, developmental stage, and status.

### 3.3. Metabolites as Markers of Drought Stress

Various abiotic stressors are known to affect the profiles of plant metabolites [113]. Indeed, the process of metabolic adjustment, i.e., accumulation of osmotically active and metabolically neutral solutes, such as different sugars, amino acids (predominantly proline and glycine), betaine, polyamines, and organic acids, under drought conditions is well documented [20]. Metabolic adjustment is the second step in plant adaptation to drought (after stomata closure) and is critical in maintaining the water status and physiological activity of plant cells, especially during relatively short-term drought [5,114]. To address the tissue contents of drought-protective metabolites, different methodological approaches can be employed. On the one hand, each group of metabolites can be analyzed individually (for example, analysis of betaine [115] and inositol [116] levels). On the other hand, entire profiles of primary metabolites can be addressed by comprehensive gas chromatography–mass spectrometry (GC-MS)-based hyphenated techniques, giving access to relative [117,118] and absolute [119] amounts of individual analytes. For a complete understanding of plant response to drought, an analysis of plant hormones and secondary metabolites can be equally essential. In this regard, Ahmed et al. reported upregulation of phenolic metabolites in the leaves of *Gossypium barbadense* L. under water deficit conditions [120], and Ma et al. demonstrated a drought-related increase of the expression levels of flavonoid genes and upregulation of leaf flavonoids in *Triticum aestivum* [121].

It is important to mention that accumulation of sugars in the background of reactive oxygen species (ROS) overproduction (usually accompanying plant response to drought) might result in enhanced formation of reactive carbonyl compounds (RCCs) and glycation of plant proteins [72,122], similar to the mechanism recently reported to occur under plant aging [123]. Additional in vitro experiments with peptide and protein models showed formation of various glycoxidative modifications of lysyl and arginyl residues [124,125,126,127] prospectively, with an impact on pro-inflammatory properties of glycated proteins [128]. Hence, these modifications might affect nutritional properties of plant-derived foods. Moreover, the processes of DNA damage and reparation (associated also with the PARP/PARG system [69]) can impact protein glycation as well [129,130].

Remarkably, metabolic adjustment in different plants has both common and species-specific features. Thus, some osmoprotective metabolites, like glycine betaine, are specific for certain plant species, e.g., sugar beet (*Beta vulgaris*), spinach (*Spinacia oleracea*), and barley (*Hordeum vulgare*) [131], while increased proline content, which is apparently a crucial and the most conserved response to drought, is characteristic for a wide range of plants [132]. Obviously, such metabolites can be used as nonspecific and species-specific markers of drought stress. It is important to remember that metabolic adjustment is efficient only on a relatively short time scale, whereas when drought persists for longer times, increased accumulation of compatible solutes can be energy- and resource-intensive for the plant. In cases of severe stress, when soil water content is largely depleted, metabolic adjustment may have only a small effect on water uptake, or even be detrimental by taking too many resources from the plants [18,133].

### 3.4. Protective Proteins as Markers of Drought Stress

Underlying the long-term adaptation of plant organisms to drought is a pronounced increase in the expression of drought-specific genes, such as *Solanum tuberosum* DS2 (StDS2) [134], late embryogenesis abundant (LEA) [135]. Accordingly, biosynthesis of a broad array of drought-protective proteins, predominantly chaperones, LEA proteins, and enzymes of antioxidant defense (referred to below in detail), is upregulated. Chaperones form the group of proteins involved in the formation and maintenance of the native protein structure [136], mostly represented by so-called heat shock proteins: the ubiquitous polypeptides, originally described with respect to a heat shock response, but actually involved in an array of stress adaptation responses [137]. Currently, special attention is being paid to the role of heat shock proteins in drought tolerance [138]. Xiang et al. found that overexpression of the heat shock protein Osnsp50.2 in rice leaf reduced water loss and increased resistance of plants to drought-related osmotic stress [139]. It was also shown that increased expression of chaperone-like proteins ERD10 and ERD14 in *A. thaliana* cells impacted the prevention of luciferase, alcohol dehydrogenase, and citrate synthase inactivation in firefly [140].

LEA proteins represent another class of polypeptides involved in adaptation to water deficiency. These proteins were discovered more than 35 years ago in a study of embryogenesis and germination of cotton seeds [141]. The key feature of the LEA proteins underlying their drought-protective properties is their high hydrophilicity [142]. These molecules are known to prevent mechanical damage of mitochondria, chloroplasts, and other cellular structures by forming a membrane-protecting shield, thereby preventing peroxidation of membrane lipids [111,143]. The constitutive expression level of LEA proteins can be considered as a marker of drought resistance. Thus, it was shown that more LEA genes are overexpressed in drought-resistant *Gossypium tomentosum* cultivars than drought-sensitive ones [144].

### 3.5. Oxidative Stress Associated with Drought

Water deficiency results in a disturbance of the balance between ROS (and RNS) generation and detoxification, triggering oxidative stress, and upregulation of ROS production under drought conditions is well documented (comprehensively reviewed by de Carvalho et al. [145]). Due to their high reactivity, ROS are extremely toxic and can damage proteins, lipids, and nucleic acids [146]. Under persistent oxidative stress, this damage can become irreversible and might lead to cell death. Indeed, excessive ROS production is a central process in response to infection (killing the intruder cells or tissues surrounding it).

Although ROS as singlet oxygen can be produced by the energy transfer from triplet chlorophyll to molecular oxygen [147] (Figure 3A), the main reason underlying overproduction of cellular ROS in response to plant dehydration is the overload of electron transport chains in chloroplasts and mitochondria due to overproduction of reduced forms of nucleotides [145,148]. Indeed, even under normal conditions, light reactions of photosynthesis are associated with continuous ROS production [149], and PS II is the main contributor to chloroplast photosynthesis [150]. The superoxide anion radical (O_2_^−^) is formed on the electron acceptor side of PS II by electron leakage into molecular oxygen (Figure 3A). Due to the drought-related stomata closure and overload of electron-transport chains, the rate of this process is essentially increased [151]. The formed O_2_^−^can be dismutated to hydrogen peroxide (H_2_O_2_), which can further yield highly toxic hydroxide radical (OH^·^), for example, by the Fenton reaction in the presence of certain transition metal ions [152]. On the PS II donor side, incomplete water oxidation also leads to H_2_O_2_ production. Dehydration affects the function of PS II, resulting in higher production of H_2_O_2_ and faster transformation into OH radical [145,150].

Another important source of ROS in chloroplasts is the Mehler reaction (Figure 3A), i.e., partial reduction of O_2_ to O_2_^−^ (with subsequent formation of H_2_O_2_) by components of the PS I–Fe-S centers and reduced ferredoxin and thioredoxin. Under stress conditions, reactions of the Calvin cycle are inhibited by lack of CO_2_ due to the stomata closure. This situation provokes an overreduction of the chloroplastic electron transport chain, which results in a higher leakage of electrons to O_2_ in the Mehler reaction [145]. Importantly, the deficit of CO_2_ might result in enhancement of H_2_O_2_ production in peroxisomes, with photorespiration contributing over 70% of the total H_2_O_2_ production in C3 plants subjected to drought stress [153].

Mitochondria also can represent an important source of stress-related excess ROS. Normally, approximately 1–2% of the oxygen consumed by plant mitochondria is converted to O2^−^ and H_2_O_2_. Underlying this increase in ROS production are complexes I and III of the mitochondrial electron transport chain, which can act as electron donors for molecular oxygen and enhance the generation of O_2_^−^ and H_2_O_2_ [148]. It is assumed that the excess NADH produced during glycine oxidation in the photorespiratory pathway results in an overload of the mitochondrial electron transport chain [154]. Interestingly, the activity of alternative oxidase, and probably rotenone-insensitive NAD(P)H-dehydrogenase, is involved in detoxification of ROS under these conditions and contribute to plant drought tolerance [148].

In general, ROS production correlates well with the severity of drought stress [145]. This allows the use of some compounds associated with oxidative stress as biochemical markers of drought. ROS readily attack double bonds in polyunsaturated fatty acids, resulting in the formation of lipid hydroperoxides [155]. Consequently, shorter and reactive carbonyl products result from their breakdown, such as, e.g., malondialdehyde, known as a reliable marker of lipid oxidative damage [156,157]. The contents of these compounds increase in plant leaves under stress conditions and can be used as drought stress markers. Similarly, H_2_O_2_ tissue contents are often used for estimations of drought stress severity in plants, as this molecule represents the most stable and easily measurable form of ROS [158].

The mechanisms of plant drought tolerance necessarily include pathways that reduce ROS content in stressed cells. The most efficient antioxidant defense relies on the activities of specific antioxidant enzymes (Figure 3B). The enzymes of the ascorbate–glutathione cycle play a central role in detoxification of H_2_O_2_ under drought stress conditions [5,159]. Ascorbate peroxidase, the key antioxidant enzyme neutralizing H_2_O_2_ in plant cells, relies on ascorbic acid as a donor of electrons [160]. The resulting dehydroascorbate can be regenerated (i.e., reduced to monodehydroascorbate) by the reaction with NADPH catalyzed by monodehydroascorbate reductase [161]. The formed toxic monodehydroascorbate is rapidly reduced to ascorbic acid by dehydroascorbate reductase, parallel to the oxidation of glutathione to glutathione disulfide (GSSG). The subsequent regeneration of glutathione (GSH) is catalyzed by glutathione reductase, which plays a key role in maintaining the pool of reduced glutathione required for survival under stress conditions [161,162].

The ratio of reduced to oxidized forms of ascorbate and glutathione is crucial for maintaining a favorable redox status of living cells, being an informative indicator of plant stress adaptation capacity [162]. Therefore, addressing the expression or activity of antioxidant enzymes may be important in screening different plant species and cultivars for drought tolerance. The enzymes of the ascorbate–glutathione cycle were recently considered as targets for the engineering of transgenic stress-resistant plants [163].

## 4. Conclusions

The comprehensive literature survey clearly demonstrates the importance of an appropriate experimental design of reversible stress induction under reproducible and long-term stable laboratory conditions. Currently, PEG-induced drought stress models in particular are state of the art. Stress characterization methods include a set of standards but also species-specific small-molecule metabolites and enzymes indicative of the elucidation of drought tolerance mechanisms in plants. In this context, multiple modifications of the drought model experimental setups allow monitoring different aspects of plant functional states, in agreement with specific objectives. Currently, the progress of studies focused on improving plant drought resistance is associated with molecular biology and omics techniques, in an effort to eventually understand and genetically or chemically influence plant responses to drought periods.

**Table 1 ijms-19-04089-t001:** Overview of drought stress model setups.

Species	Drought Stress Model	Osmotically Active Agent	Age of Plant	Duration of Stress	Reference
*Arabidopsis thaliana* L.	Agar system	50, 300 mmol/L mannitol	7 days	2 weeks	[164]
*Arabidopsis thaliana* L.	Agar system	100, 200, 300 mmol/L mannitol	8 days	1 day	[165]
*Arabidopsis thaliana* L.	Agar system	17% PEG8000	2 weeks	3 days	[100]
*Lemna minor* L.	Hydroponic system (Microtiter plate formate possible)	PEG6000 or 8000 Variable conc.	Adult	24 h	[68]
*Hordeum vulgare* L.	Soil system	No	Adult	Every 15 days until physiological maturity	[166]
*Zea mays* L.	Soil system	No	Adult	Every 15 days until physiological maturity	[166]
*Zea mays* L.	Hydroponic system	15% PEG6000	5 weeks	24 h	[167]
*Populuseuphratica*	Soil system	No	2 months	0, 4, 8 24, 48, 96 h	[168]
*Solanum tuberosum* L.	Agar system	Sorbitol (0.1, 0.2, 0.3, and 0.4 m) and PEG8000 (0%, 4.8%, and 9.6%)	2 weeks	3 weeks	[169]
*Lolium perenne* L.	Hydroponic system	10, 20% PEG6000	1 week	4 weeks	[170]
*Solanum lycopersicum* L.	Hydroponic system	15% PEG8000	25 days	0, 3, 6, 24, 48 h	[171]
*Medicago sativa*	Hydroponic system	15% PEG6000	28 days	24 h	[172]
*Pistacia lentiscus*	Soil system	5, 10, 15, 20, 25% PEG6000	1,5 months	20, 23 days	[173]
*Brachypodiumdistachyon*	Soil system	No	Vegetativestage	4, 8, 12 days	[174]
*Transgenicplum “Claudia verde”*	Soil system	No	8 weeks	7, 15 days	[175]
*Stipapurpurea*	Soil system	No	Trefoilstage (about 3 weeks’ growth)	7, 15 days	[176]
*Saccharum* spp.	Soil system	No	2 months	17 days	[177]
*Hordeumvulgare* L.	Hydroponic system	20% PEG6000	31 days	9 days	[178]
*Brassica campestris* ssp.	Hydroponic system	60, 120%PEG6000	34 days	7 days	[179]
*Oryzasativa* L.	Soil system	No	Reproductive stage	–	[180]
*Cucumissativus* L.	Hydroponic system	2% PEG6000	2 weeks	7 days	[181]

**Table 2 ijms-19-04089-t002:** Markers of drought stress in plants.

Parameter	Growth Model	Plant Object	Method	Reference
**Physiological Markers**				
Leaf water potential (MPa)	Soil	Cotton (*Gossypium hirsutum* L.)	Pressure chamber technique	[182]
Relative water content (RWC; %)	Soil	Potato (*Solanum tuberosum* L.)	RWC (%) = [(FW − DW)/(SW − DW)] × 100, where FW, DW, and SW are fresh, dry, and saturated (turgid)weights of leaf tissues, respectively	[183]
Stomatal conductance	Soil	Tomato (*Lycopersicon esculentum* Mill.)	Abaxial stomatal conductance measurement with diffusion porometer (AP4, Delta-T, Cambridge, UK)	[90]
Photosynthetic parameters (chlorophyll content and PSII activity)	Soil	Barley (*Hordeum vulgare* L.)	Determination of leaf chlorophyll using chlorophyll meter (SPAD-502, Minolta, Japan); measurement of chlorophyll fluorescence with portable fluorescence spectrometer (Handy PEA, Hansatech Instruments, Norfolk, UK) Fluorescence value Fv/Fm represents maximum quantum yield of PSII: Fv = Fm−Fo	[96]
**Biochemical Markers**				
Phytohormones	Soil	Clover (*Trifolium subterraneum* L.)	ABA analysis in xylem sap by ELISA	[184]
	Soil	Wheat (*Triticum aestivum* L.)	ABA analysis by HPLC	[185]
Metabolites	Soil	*Triticum* spp.	LMW drought stress–responsive metabolites in root and leaf samples of 7 wild and domesticated wheatrelatives revealed by GC-MS based comparative metabolomicsapproach	[186]
Protective proteins	Soil	Rice (*Oryza sativa* L.)	Expression pattern analysis of OsHSP50.2, an HSP90 family gene	[139]
	Soil	Cotton (*Gossypium tomentosum, Gossypium hirsutum*)	LEA gene expression analysis and profiling	[144]
ROS and antioxidant enzymes	Water culture + PEG6000	Wheat genotypes	–	[187]

PS, photosystem II; ABA, abscisic acid; LMW, low molecular weight; LEA, late embryogenesis abundant.

## Figures and Tables

**Figure 1 ijms-19-04089-f001:**
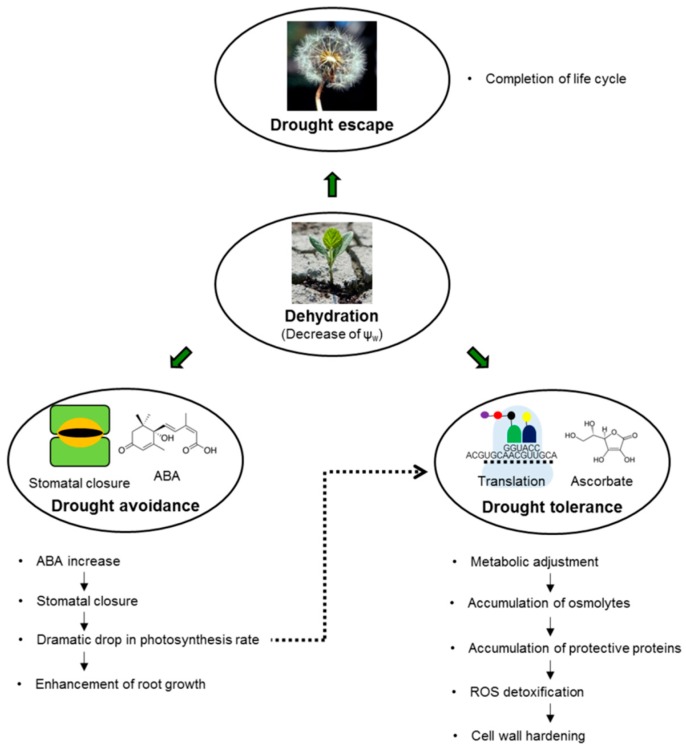
The main drought resistance strategies employed by plants to counter water deficit periods (drought escape, drought avoidance, and drought tolerance) and the main steps of the plant response to dehydration.

**Figure 2 ijms-19-04089-f002:**
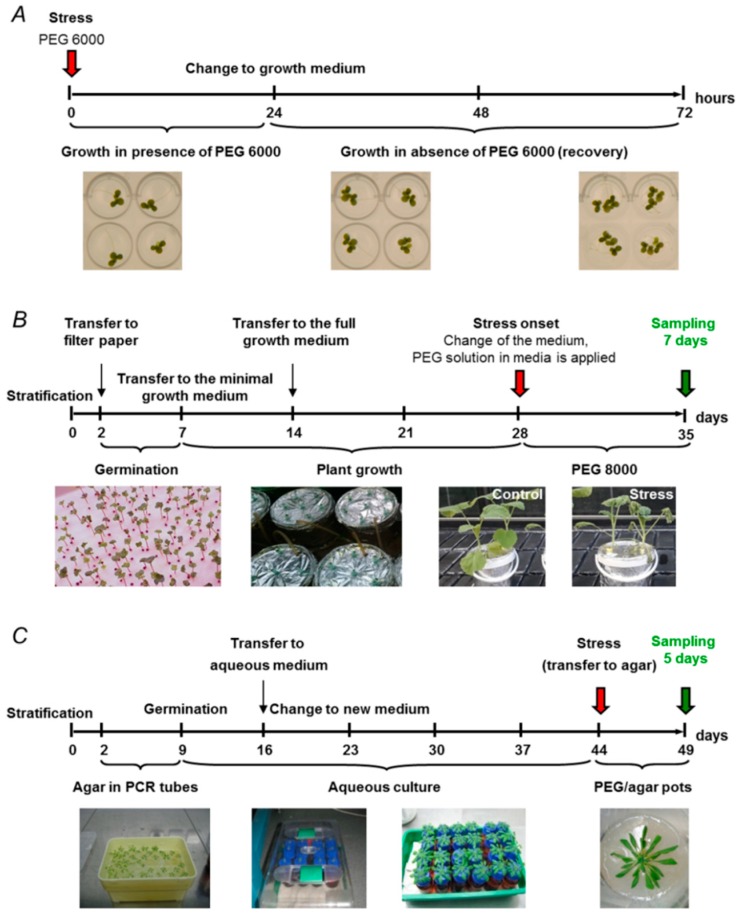
Experimental drought models based on osmotic stress and established by supplementation of growth medium with polyethylene glycol (PEG): (**A**) *Lemna minor* model, established with aqueous growth medium supplemented with PEG6000 ([68]); (**B**) *Brassica napus* model, established with aerated aqueous culture supplemented with PEG8000; and (**C**) agar-based PEG infusion *Arabidopsis thaliana* model, established by overlaying solidified agar medium with PEG8000 solution for five days.

**Figure 3 ijms-19-04089-f003:**
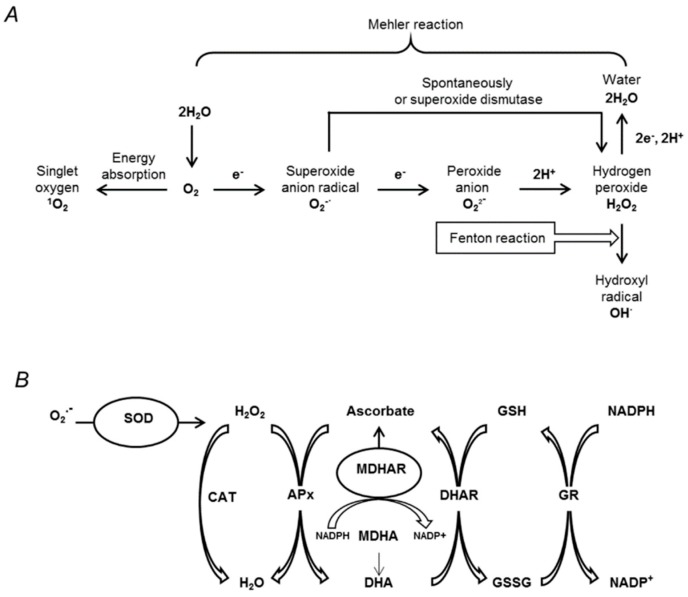
(**A**) The main pathways of reactive oxygen species (ROS) generation in plants and (**B**) the major pathways of plant enzymatic antioxidant defense. SOD, superoxide dismutase; CAT, catalase; APx, ascorbate peroxidase; MDHA, monodehydroascorbate; MDHAR, monodehydroascorbate reductase; DHA, dehydroascorbate; DHAR, dehydroascorbate reductase; GSH, reduced glutathione; GSSG, oxidized glutathione; GR, glutathione reductase.

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
