# Peer review of "Methodology of Drought Stress Research: Experimental Setup and Physiological Characterization"

_ijms, 2018, doi:10.3390/ijms19124089_

Round 1
Reviewer 1 Report
The submitted manuscript entitled “Modeling of drought stress: experimental setup and physiological characterization” reviewed experimental setup for physiological characterization of plant exposed to drought stress. The authors overviewed three drought models based on culturing plants in soil, hydroponic, and agar culture. The authors also summarized physiological and biochemical characterization of plants exposed to drought. This manuscript is within the scope of International Journal of Molecular Sciences and I believe that this manuscript is worthful for publication in the journal. I would provide a bit comments as below for further improvement of this manuscript.
Line 88: “Drought resistance” was not defined in this manuscript. “Drought resistance” may include all the strategies employed by plants exposed to drought stress: drought escape, drought avoidance, and drought tolerance, as shown in Figure 1. Please consider presenting what “drought resistance” implies here.
Line 137: “w” of “Ψw” should be a subscript text.
Line 163: The authors revealed “the roots of both wild type and experimental plants grow in the same medium and have, therefore, the same root Ψw”, however, I am wondering if it is correct. If the plants grown in the same medium, the soil Ψw will be the same for each plant. However, root Ψw of each plant will not be predictable. The paper by Verslues et al. (2006), which had been cited by the authors, revealed “the roots of both genotypes will grow into the same soil and be exposed to the same Ψw even if one genotype uses water more quickly than the other.” This may imply that soil Ψw will be the same but root Ψw can be different. Please check.
Line 213: “PARP” should be defined here as “poly (ADP-ribose) polymerase (PARP)”.
Line 221: A brief explanation about “2D-photodocumentation visualization system” should be added.
Line 462: PS II had been already defined at line 342. “Photosystem II” may not be required here.
Line 490: “(B)” should be inserted after “plant enzymatic antioxidant defense.”
Paper by Ito et al. (2006) was listed twice; numbered 38 and 41. Please delete one of them.
The authors of the paper numbered 181 were not indicated. Please put in.
Author Response
We thank the reviewer for the thoughtful review and highly appreciate the valuable comments and suggestions to improve the manuscript. Following these advices we performed all required changes in corresponding sections, as indicated in the following rebuttal addressing each aspect.
Reviewer: 1
Remarks
Remark 1: Line 88: “Drought resistance” was not defined in this manuscript. “Drought resistance” may include all the strategies employed by plants exposed to drought stress: drought escape, drought avoidance, and drought tolerance, as shown in Figure 1. Please consider presenting what “drought resistance” implies here.
Answer: Yes, we agree, this need to be referred. We provide this now in text:
“Generally, all these three strategies impact on the development of the state, known as drought resistance, which can be defined as ability to maintain favorable water balance and turgidity under drought conditions” (Lines 79-81)
Remark 2: Line 137: “w” of “Ψw” should be a subscript text.
Answer: Changed accordingly (Line 154-156)
Remark 3: Line 163: The authors revealed “the roots of both wild type and experimental plants grow in the same medium and have, therefore, the same root Ψw”, however, I am wondering if it is correct. If the plants grown in the same medium, the soil Ψw will be the same for each plant. However, root Ψw of each plant will not be predictable. The paper by Verslues et al. (2006), which had been cited by the authors, revealed “the roots of both genotypes will grow into the same soil and be exposed to the same Ψw even if one genotype uses water more quickly than the other.” This may imply that soil Ψw will be the same but root Ψw can be different. Please check.
Answer: Yes of course, we change the text accordingly:
“In this case, both reference or wild type and experimental or mutant plants would grow in the same medium and, therefore, exposed to the same soil Ψw if they are planted in a suitable scheme and position.” (Line 179-182)
Remark 4: Line 213: “PARP” should be defined here as “poly (ADP-ribose) polymerase (PARP)”.
Answer: Changed accordingly (Line 253)
Remark 5: Line 221: A brief explanation about “2D-photodocumentation visualization system” should be added.
Answer: Yes, this photodocumentation system registers an increase of plant leaf area accompanying its development. Thus, attenuation of growth inhibition can be registered by changes in kinetics of this increase. We changed the text accordingly:
“…protective effect is assessed by attenuation of growth inhibition via measurement of leaf peak area increase by means of a 2D-photodocumentation visualization system” (Lines 264-265)
Remark 6: Line 462: PS II had been already defined at line 342. “Photosystem II” may not be required here.
Answer: Changed accordingly (Line 532)
Remark 7: Line 490: “(B)” should be inserted after “plant enzymatic antioxidant defense.
Answer: Changed accordingly (Line 586)
Remark 8: Paper by Ito et al. (2006) was listed twice; numbered 38 and 41. Please delete one of them.
Answer: Changed accordingly, literature list is updated
Remark 9: The authors of the paper numbered 181 were not indicated. Please put in.
Answer: Changed accordingly, literature list is updated
Reviewer 2 Report
This article aims at reviewing the diverse experimental setups for imposing water stress and the various biochemical markers used for drought response characterization. The article is well-written and is interesting thanks to its critical analysis of the methods which have been used in the literature. Although reviews on methods towards the quantification of drought resistance already exist, this one has the particularity to combine how the experiments should be performed according to the associated research questions and how to kinetically follow plant response to drought.
However, here are some comments which could help improving the manuscript:
1- Title
The words “model” or “modeling” is used along the whole manuscript. It is hazardous tu use this term in the title because a large audience could expect to read a review on a mechanistic / statistic modeling of plant response to drought including a prediction dimension…and could thus be disappointed. You should use a title which better reflect your reviewing work.
2- Abstract
The abstract should more highlight the added value of the present paper instead of giving some well-known aspects of plant response to drought. Please indicate more clearly the novelty of your work.
3- Introduction
- Lines 48 – 54: a list of plant responses to drought is given but a chronological aspect is missing
- Lines 55 – 59: you mention water deficit and water stress, but you should better define to what exactly refers the first concept and what is different in the second.
- Lines 75 – 87: please take into account the paper from Muller et al., 2011 which demonstrate that organ expansion (as a major C sink) is affected earlier and more intensively than photosynthesis (C source) and metabolism.
- Line 88: you mention “plant resistance” but you’ve never defined it, especially in the figure 1.
- Line 103: change “drought-relates” into “drought-related”
- Line 108: add a comma after “experimental conditions”
4- Experimental models of drought stress
- Line 117: you never mention the inert substrates: do they fall into the soil-based experimentation?
- Line 136: please precise what you are referring to under the expression “multiple important aspects of plant adaptation”
- Lines 151 and after: I don’t understand why the air treatment is in the “soil-based drought models” section
- Line 156: your opinion is that the air treatment is a very reliable and reproducible method to evaluate drought stress, but you never mention that it highly depends on the air VPD.
- Line 174: you mention that the easiest way to reduce the water potential is by decreasing the level of nutrient solution, but it means that the water AND nutrients are simultaneously decreased. What about decreasing the level of water without decreasing the levels of nutrients?
- Lines 177-178: I don’t understand why you mention nitrate in priority over the other nutrients and why you associate these decreases with strong up-regulation of sugars in leaves. The first is linked to the ion availability in the soil while the second is related to the changes in plat functioning.
- Line 213: define PARP
- Figure 2: in the legend, precise that agar-based PEG infusion is (C). For the (C), PEG/agar pots could be illustrated with a picture taken from the side of the pot (and not a somital view).
- Lines 291-296: When you summarize the different experimental setups, please be more critical towards the use of agar-based models, which cannot allow to directly transpose drought effects in the field / ecosystem due to the extreme simplification of the model: the water content gradient is not mimicked, as well as soil heterogeneity in terms of water holding capacity for some root segments…
5- Physiological and biochemical characterization of drought stress
- Line 307: Before the description of plant responses, I suggest to introduce a section explaining how the drought stress is perceived by the plant
- Lines 312: Values indicating at which water potential level plants are considered to be submitted to water stress are welcome, however please precise that these values are species-dependent.
- Lines 396-399: you could give examples to illustrate and sustain this important statement
- Lines 426-447: I would rather begin the physiological and biochemical characterization section by the changes in phytohormones patterns, as it seems to be hormones that are determinant in the subsequent plant responses to drought
- I think a table summarizing the physiological and biochemical markers for drought stress characterization and the associated protocols (by references) would be a high added value to the article.
Author Response
We thank the reviewer for the thoughtful review and highly appreciate the valuable comments and suggestions to improve the manuscript. Following these advices we performed all required changes in corresponding sections, as indicated in the following rebuttal addressing each aspect.
Reviewer: 2
Remarks
Remark 1: 1- Title. The words “model” or “modeling” is used along the whole manuscript. It is hazardous to use this term in the title because a large audience could expect to read a review on a mechanistic / statistic modeling of plant response to drought including a prediction dimension…and could thus be disappointed. You should use a title which better reflect your reviewing work.
Answer: We agree with the Reviewer, and change the title:
“Methodology of drought stress research: experimental setup and physiological characterization”
Remark 2: 2- Abstract. The abstract should more highlight the added value of the present paper instead of giving some well-known aspects of plant response to drought. Please indicate more clearly the novelty of your work.
Answer: We agree with the Reviewer, and re-write the second half of the abstract:
“…Therefore, in this review we comprehensively address currently available models of drought stress, based on culturing plants in soil, hydroponic or agar culture. Thereby, we critically discuss advantages and limitations of each design. We also address the methodology of drought stress characterization and discuss it in the context of real experimental approaches. Further, we highlight the trends of the methodological development in the drought stress research, i.e. complementation of conventional tests with quantification of phytohormones and reactive oxygen species (ROS), measurement of antioxidant enzyme activities, as well as comprehensive profiling of transcriptome, proteome and metabolome” (Lines 25 - 32)
Remark 3: Introduction. Lines 48 – 54: a list of plant responses to drought is given but a chronological aspect is missing
Answer: The time dimension is implemented:
“First, drought compromises stomata function, impairs gas exchange, and leads to over-production of reactive oxygen species (ROS) and development of oxidative stress [ ]. Secondly, water deficit inhibits cell division, expansion of leaf surface, growth of stem and proliferation of root cells [ ].” (Lines 50-53)
Remark 4: Introduction. Lines 55 – 59: you mention water deficit and water stress, but you should better define to what exactly refers the first concept and what is different in the second
Answer: Of course, water deficit corresponds to environment, and water stress – to change of water potential in plant. We adjust the text in agreement with this:
“At the quantitative level, water deficit in the environment can be characterized by a decrease of soil water potential (Ψw) [18]. According to the Van't Hoff equation, it indicates a decrease in free energy of substrate water that makes water uptake from the medium under these conditions thermodynamically unfavorable and loss of water by the plant more probable. Ψw of 0 to -0.3 MPa are characteristic for well-watered plants, whereas the values below -0.4 MPa correspond to moderate water stress, and potentials of -1.5 to -2.0 MPa represent severe stress and permanent loss of turgor” (Lines 55-61).
Remark 5: Introduction. Lines 75 – 87: please take into account the paper from Muller et al., 2011 which demonstrate that organ expansion (as a major C sink) is affected earlier and more intensively than photosynthesis (C source) and metabolism.
Answer: Changed accordingly (Lines 93-96)
Remark 6: Introduction. Line 88: you mention “plant resistance” but you’ve never defined it, especially in the figure 1.
Answer: Please see our answer to the Remark 1 of the Reviewer 1
Remark 7: Introduction. Line 103: change “drought-relates” into “drought-related”
Answer: Changed accordingly (Line 121)
Remark 8: Introduction. Line 108: add a comma after “experimental conditions”
Answer: Changed accordingly (Line 126)
Remark 9: Line 117: you never mention the inert substrates: do they fall into the soil-based experimentation?
Answer: The information is provided:
“It is important to mention the setups, relying on inert substrate, such as vermiculite or perlite, as soil substitutes. The advantages of this approach is that the roots of experimental plants can be pulled out easily and without damage to investigate drought-related changes in water potential [46 ] or oxidative and metabolic responses [47] at the root level. Inert substrates are suitable for studying the effects of drought in legume-rhizobial nodule symbiosis [48]. On the other hand, their certain disadvantage is that watering unlike soil culture, is carried out not with water, but with a nutrient solution, so the impact of drought by cessation of watering plants is accompanied by the appearance of another stress factor, namely, the deficiency of mineral elements.” (Lines 186-193)
Remark 10: Line 136: please precise what you are referring to under the expression “multiple important aspects of plant adaptation”
Answer: The effects, which might be missed in soil models are specified:
“many important aspects of plant drought tolerance and adaptation to low Ψw, like, for example, accumulation of osmoprotective proteins and hardening of cell walls, can be overlook” (Lines 154-156)
Remark 11: Lines 151 and after: I don’t understand why the air treatment is in the “soil-based drought models” section
Answer: Sorry for this thinking mistake. We have put this model to hydroponic section, and changed the text a bit:
“To simulate severe dehydration, plant roots can be left under air for up to eight hours [38]. Thereby, the severity of simulated drought can be defined by the duration and repetitions of such dehydration procedure” (Line 202-204)
Remark 12: Line 156: your opinion is that the air treatment is a very reliable and reproducible method to evaluate drought stress, but you never mention that it highly depends on the air VPD.
Answer: We completely agree with the Reviewer. Therefore, we add an appropriate remark to the text:
“When using this approach, however, one needs to keep in mind, that dehydration degree and kinetics would strongly depend on air humidity” (Line 207-208)
Remark 13: Line 174: you mention that the easiest way to reduce the water potential is by decreasing the level of nutrient solution, but it means that the water AND nutrients are simultaneously decreased. What about decreasing the level of water without decreasing the levels of nutrients?
Answer: Unfortunately, here we have expressed our idea not in the best way, and it caused miss-understanding: we mean here the level of liquid in the pots. The changes in text are done:
“The easiest way to reduce the Ψw of growth medium assumes decreasing its level in pots and partial exposure of roots to air, as was shown for lettuce by Koyama et al”(Lines 200-202)
Remark 14: Lines 177-178: I don’t understand why you mention nitrate in priority over the other nutrients and why you associate these decreases with strong up-regulation of sugars in leaves. The first is linked to the ion availability in the soil while the second is related to the changes in plat functioning.
Answer: Yes, it was not logic from our side – we remove these two sentences.
Remark 15: Line 213: define PARP
Answer: Changed accordingly (Line 253)
Remark 16: Figure 2: in the legend, precise that agar-based PEG infusion is (C). For the (C), PEG/agar pots could be illustrated with a picture taken from the side of the pot (and not a somital view).
Answer: The panel reference is added. A somital view of the pot is hardly possible, as the wall is not completely optically clear – we tried earlier, but, unfortunately, failed.
Remark 17: Lines 291-296: When you summarize the different experimental setups, please be more critical towards the use of agar-based models, which cannot allow to directly transpose drought effects in the field / ecosystem due to the extreme simplification of the model: the water content gradient is not mimicked, as well as soil heterogeneity in terms of water holding capacity for some root segments…
Answer: This is incorporated in the discussion of model comparison:
“It is necessary to keep in mind, however, that this setup doesn’t allow a direct extrapolation of drought effects to the field or ecosystem due to high simplicity of the model, which doesn’t consider water gradient in soil and heterogeneity in terms of water holding capacity” (Lines 350-353)
Remark 18: Line 307: Before the description of plant responses, I suggest to introduce a section explaining how the drought stress is perceived by the plant
Answer: The requested part is added:
“Thus, ideally, selection of the markers needs to consider all steps of drought response, starting from drought perception. It is assumed, that the drought is recognized by roots, which send a chemical message to the shoot [ ]. Abscisic acid (ABA) plays the key role in this signaling [ ]. This effector is synthesized in response to hydraulic signal in vascular tissues and further transported to leaf epidermis cells. Resulting stomata closure results in suppression of xylem transport, decrease of turgor and root growth arrest [ ].” (Lines 365-370)
Remark 19: Lines 312: Values indicating at which water potential level plants are considered to be submitted to water stress are welcome, however please precise that these values are species-dependent.
Answer: This consideration is incorporated in text:
“Although critical values of tissue water potentials are species-specific…” (Lines 376-377)
Remark 20: Lines 396-399: you could give examples to illustrate and sustain this important statement
Answer: The sentence is modified accordingly:
“Thus, some osmoprotective metabolites, like glycine betaine, are specific for certain plant species, e.g. sugar beet (Beta vulgaris), spinach (Spinacia oleracea), and barley (Hordeum vulgare) [ ]…” (Lines 482-485)
Remark 21: Lines 426-447: I would rather begin the physiological and biochemical characterization section by the changes in phytohormones patterns, as it seems to be hormones that are determinant in the subsequent plant responses to drought
Answer: We agree with the reviewer, and put phytohormone section before all other biochemical markers. But we would keep physiological parameters as the first. The logic for this – these marker are usually measured as the first, before phytohormones or oxidative stress markers.
Remark 22: I think a table summarizing the physiological and biochemical markers for drought stress characterization and the associated protocols (by references) would be a high added value to the article.
Answer: The table is prepared – see Table 2